# Antibiotic Resistance in Non-Typhoidal *Salmonella enterica* Strains Isolated from Chicken Meat in Indonesia

**DOI:** 10.3390/pathogens11050543

**Published:** 2022-05-04

**Authors:** Minori Takaichi, Kayo Osawa, Ryohei Nomoto, Noriko Nakanishi, Masanori Kameoka, Makiko Miura, Katsumi Shigemura, Shohiro Kinoshita, Koichi Kitagawa, Atsushi Uda, Takayuki Miyara, Ni Made Mertaniasih, Usman Hadi, Dadik Raharjo, Ratna Yulistiani, Masato Fujisawa, Kuntaman Kuntaman, Toshiro Shirakawa

**Affiliations:** 1Department of Public Health, Kobe University Graduate School of Health Sciences, Kobe 654-0142, Japan; minori.wanko.2804@gmail.com (M.T.); mkameoka@port.kobe-u.ac.jp (M.K.); yutoshunta@gmail.com (K.S.); 2Department of Medical Technology, Kobe Tokiwa University, Kobe 653-0838, Japan; m-miura@kobe-tokiwa.ac.jp; 3Department of Infectious Diseases, Kobe Institute of Health, Kobe 650-0046, Japan; ryohei_nomoto@office.city.kobe.lg.jp (R.N.); noriko_nakanishi@office.city.kobe.lg.jp (N.N.); 4Division of Urology, Department of Organ Therapeutics, Faculty of Medicine, Kobe University Graduate School of Medicine, Kobe 650-0017, Japan; masato@med.kobe-u.ac.jp (M.F.); toshiro@med.kobe-u.ac.jp (T.S.); 5Division of Advanced Medical Science, Kobe University Graduate School of Science, Technology and Innovation, Kobe 650-0017, Japan; kino@med.kobe-u.ac.jp (S.K.); ko1.kitgwa@gmail.com (K.K.); 6Department of Infection Control and Prevention, Kobe University Hospital, Kobe 650-0017, Japan; a-uda@umin.ac.jp (A.U.); miyarat@med.kobe-u.ac.jp (T.M.); 7Department of Pharmacy, Kobe University Hospital, Kobe 650-0017, Japan; 8Department of Microbiology, Faculty of Medicine, Airlangga University, Surabaya 60132, Indonesia; nmademertaniasih@gmail.com (N.M.M.); usmanhadi@sby.centrin.net.id (U.H.); 9Institute of Tropical Disease, Airlangga University, Surabaya 60286, Indonesia; dadik.tdc@gmail.com (D.R.); kuntaman@fk.unair.ac.id (K.K.); 10Department of Veterinary Public Health, Faculty of Veterinary Medicine, Airlangga University, Surabaya 60115, Indonesia; 11Department of Food Technology, Faculty of Engineering, Universitas Pembangunan Nasional Veteran Jawa Timur, Surabaya 60294, Indonesia; ratna.tp@upnjatim.ac.id

**Keywords:** non-typhoidal *Salmonella enterica* (NTS), Indonesia, antimicrobial resistance, virulence factors, chicken

## Abstract

The increase in antibiotic resistance in non-typhoidal *Salmonella* enterica (NTS) has been confirmed in Indonesia by this study. We confirmed the virulence genes and antimicrobial susceptibilities of clinical NTS (*n* = 50) isolated from chicken meat in Indonesia and also detected antimicrobial resistance genes. Of 50 strains, 30 (60%) were non-susceptible to nalidixic acid (NA) and all of them had amino acid mutations in *gyrA*. Among 27 tetracycline (TC) non-susceptible strains, 22 (81.5%) had *tetA* and/or *tetB*. The non-susceptibility rates to ampicillin, gentamicin or kanamycin were lower than that of NA or TC, but the prevalence of bla*_TEM_* or *aadA* was high. Non-susceptible strains showed a high prevalence of virulence genes compared with the susceptible strains (*tcfA*, *p* = 0.014; *cdtB*, *p* < 0.001; *sfbA*, *p* < 0.001; *fimA*, *p* = 0.002). *S*. Schwarzengrund was the most prevalent serotype (23 strains, 46%) and the most frequently detected as multi-antimicrobial resistant. The prevalence of virulence genes in *S*. Schwarzengrund was significantly higher than other serotypes in *hlyE* (*p* = 0.011) and *phoP/Q* (*p* = 0.011) in addition to the genes above. In conclusion, NTS strains isolated from Indonesian chicken had a high resistance to antibiotics and many virulence factors. In particular, *S.* Schwarzengrund strains were most frequently detected as multi-antimicrobial resistant and had a high prevalence of virulence genes.

## 1. Introduction

*Salmonella enterica* subsp. *enterica* is broadly classified into typhoid *Salmonella*, such as *S. enterica* serovars Typhi and Paratyphi A, and non-typhoid *Salmonella* (NTS). More than 2600 serotypes of NTS have been identified, and many are known to cause invasive infections or enterocolitis with diarrhea in humans [1,2]. NTS can be easily acquired and spread by the consumption of contaminated foods of animal origin, including eggs, beef, dairy products and poultry [3]. The main symptoms of NTS infections are gastroenteritis, including diarrhea, sepsis, endocarditis, pulmonary infections, and intra-abdominal infections [3]. It is estimated that NTS causes 93.8 million cases of acute gastroenteritis and 15,000 deaths worldwide each year, and it is estimated that 86% of these are food-borne infections [4]. The pathogenicity of NTS is defined by a variety of factors encoded in virulence genes involved in adhesion (*fimA, agfA*), invasion (*invA, fliC, sopB*), survival and replication in macrophages (*phoP/Q, slyA*), systemic infection (*spvC, ssel*), fimbrial expression (*tcfA*), toxin production (*hlyE, cdtB*), and Mg^2+^ and iron uptake (*sfbA*) [5,6,7,8,9,10,11]. 

Ciprofloxacin (CPFX), ceftriaxone (CTRX) and azithromycin (AZM) are recommended for treatment for some patients with NTS infection [12]. However, NTS strains resistant to these antibiotics have been identified, making treatment clinically difficult [13]. The mechanisms of antimicrobial resistance of NTS are generally the production of antimicrobial inactivating enzymes (*aac(6′)*-*lb-cr*: quinolone resistance, *bla*_TEM_: ampicillin resistance, *aadA*: aminoglycoside resistance), modification of antimicrobial targets, such as *gyrA*, *gyrB*, *parC* and *parE* (quinolone resistance-determining region: QRDR) and *qnrA, qnrB* and *qnrS* (plasmid-mediated quinolone resistance: PMQR), antimicrobial efflux (*qepA*: quinolone resistance, *tetA*, *tetB, tetC* and *tetG*: tetracycline resistance), and the restriction of antimicrobial uptake [12,14,15,16].

The antimicrobial resistance of NTS has been increasing not only in humans but also in poultry in many countries, especially in Asia [17]. The rapid increase in antimicrobial-resistant NTS has become a major public health problem in both developing and developed countries [1]. Misuse and overuse of antibiotics are believed to be the main reasons for the increase in antimicrobial-resistant bacteria [18]. Since the major source of NTS infection is food of animal origin, it has been suggested that the presence of antimicrobial-resistant NTS may be transferred through the food chain to humans [19].

Indonesia is one of the countries projected to have the largest percentage increase in antimicrobial consumption by 2030 [20]. In Indonesia, where about 90% of the population is Muslim, chicken meat accounts for a high percentage of meat consumption [21]. Therefore, if the chicken meat is contaminated with antimicrobial-resistant NTS, there is a possibility that infection with antimicrobial-resistant NTS will spread in Indonesia, and this has not been fully investigated. In this study, we confirmed the virulence genes and antimicrobial susceptibilities of NTS isolated from chicken meat in Indonesia, and also detected antimicrobial-resistant genes. We additionally determined the relatedness among the strains by multilocus sequence typing (MLST).

## 2. Results

### 2.1. Serotyping

Table 1 shows the O serotype groups and serotypes determined from somatic (O) and flagella (H) antigens in 50 strains. The most common O serotype was the O4 group (28 strains, 56%). There were 2 strains in the O2 group, 3 strains in the O7 group, 11 strains in the O8 group, 3 strains in the O9 group, and 2 strains in the O3,10 group, and 1 strain in the O1,3,19 group. *S*. Schwarzengrund in the O4 group was the most common serotype (23 strains, 46%), followed by *S*. Istanbul in the O8 group (4 strains, 8%).

### 2.2. Antimicrobial Susceptibility Testing 

The results of antimicrobial susceptibility testing are shown in Table 2. Of 50 strains, 34 (68%) were non-susceptible to at least one antibiotic. In detail, the strains were non-susceptible to: ampicillin (ABPC; 13 strains, 26%), amoxicillin/clavulanate (AMPC/CVA; 3 strains, 6%), gentamicin (GM; 6 strains, 12%), kanamycin (KM; 8 strains, 16%), tetracycline (TC; 27 strains, 54%), CPFX (5 strains, 10%) and nalidixic acid (NA; 30 strains, 60%). In addition, five strains (10%) were non-susceptible to CPFX, which is recommended as a therapeutic agent, but all were susceptible to CTRX and AZM, which are also recommended. Of 34 strains, 27 strains (79.4%) were non-susceptible to two or more antibiotics. 

Twelve strains (35.3%) were non-susceptible to ABPC, KM and/or GM, TC and NA, including CPFX.

All *S.* Schwarzengrund were non-susceptible to at least one antibiotic (23 strains, 100%) and were especially non-susceptible to ABPC (11 strains, 47.8%), AMPC/CVA (1 strain, 4.3%), GM (5 strains, 21.7%), KM (7 strains, 30.4%), TC (22 strains, 95.7%), CPFX (2 strains, 8.7%) and NA (23 strains, 100%). Among 23 strains, 22 strains (95.7%) were non-susceptible with two or more antibiotics. Eleven strains (47.8%) were non-susceptible to ABPC, KM and/or GM, TC and NA, including CPFX. In the other serotypes, 11 of the 27 strains (40.7%) were non-susceptible to at least one antibiotic, and 5 strains (18.5%) were non-susceptible to two or more antibiotics, and 2 strains (7.4%) were non-susceptible to three or more antibiotics. *S*. Schwarzengrund had significantly higher non-susceptible rates than other serotypes for ABPC (*p* = 0.001), KM (*p* = 0.007), TC (*p* < 0.001) and NA (*p* < 0.001).

### 2.3. Detection of Antimicrobial Resistance Genes

The prevalence of antimicrobial resistance genes is shown in Table 3. Most of the non-susceptible strains to ABPC (12 of 13 strains, 92.3%) possessed *bla*_TEM_. Eleven strains were *S.* Schwarzengrund and one strain was *S*. Budapest. Among six non-susceptible strains to GM, four (66.7%) had *aadA* and all of them were *S.* Schwarzengrund. In 27 non-susceptible strains to TC, 22 (81.5%) had *tetA* and/or *tetB*, of which 15 (68.2%) had *tetA*, 6 (27.3%) had *tetB*, and 1 (4.5%) had *tetA and tetB,* but not *tetC* or *tetG*. Of 15 strains with *tetA*, 11 were *S.* Schwarzengrund and 4 were *S*. Istanbul. Of six strains with *tetB*, five were *S.* Schwarzengrund and one was *S*. Budapest. One strain that had *tetA and tetB was S.* Schwarzengrund. Among 30 strains non-susceptible to NA, QRDR mutations, especially *gyrA*, were detected in all strains. The S83→Y mutation in *gyrA* was the most common mutation (27 strains; 90%), including all 23 strains of *S.* Schwarzengrund. The D87→N mutation in *gyrA* was found in two strains (6.7%) which are *S.* Enteritidis. In addition, only one strain (3.3%) of the other serotype had two mutations of *gyrA* (S83→F and D87→N) with a mutation in *parC* (S81→I). PMQR was not found in these strains.

### 2.4. Detection of Genes Encoding Virulence Factors

The prevalence of genes encoding virulence factors is shown in Table 4. Of 13 genes, *invA* was confirmed in all strains (50 strains, 100%). Other genes were detected as follows: *sopB* (44 strains, 88%), *tcfA* (37 strains, 74%), *hlyE* (43 strains, 86%), *cdtB* (25 strains, 50%), *sfbA* (31strains, 62%), *agfA* (44 strains, 88%), *fimA* (32 strains, 64%), *slyA* (30 strains, 60%), and *phoP/Q* (43 strains, 86%). No strains had *sseI, fliC* or *spvC*.

Prevalence of virulence genes in non-susceptible strains compared with susceptible strains showed significant differences in *tcfA* (85.3% vs. 50%; *p* = 0.014), *cdtB* (73.5% vs. 0%; *p* < 0.001), *sfbA* (91.2% vs. 0%; *p* < 0.001) and *fimA* (79.4% vs. 31.3%; *p* = 0.002). 

All *S.* Schwarzengrund strains showed non-susceptibility to antibiotics and the prevalence of virulence genes in *S.* Schwarzengrund strains was significantly higher than other serotypes strains in *tcfA* (100% vs. 60.9%; *p* < 0.001), *hlyE* (100% vs. 74.1%; *p* = 0.011), *cdtB* (91.3% vs. 14.8%; *p* < 0.001), *sfbA* (100% vs. 29.6%; *p* < 0.001), *fimA* (91.3% vs. 40.7%; *p* < 0.001), and *phoP/Q* (100% vs. 74%; *p* = 0.011).

### 2.5. MLST

*S.* Schwarzengrund had a higher rate of non-susceptibility to antibiotics than other serotypes and all *S.* Schwarzengrund strains had the S83→Y mutation in *gyrA* and a high prevalence of virulence genes. Therefore, we determined the homology of 23 *S.* Schwarzengrund strains by MLST. They were classified into sequence type (ST) 96 (22 strains, 95.7%) while one strain (4.3%) differed only in *hisD* and was not typed (Table 5).

## 3. Discussion

Our study investigated the antimicrobial susceptibilities and genetic analysis of 50 NTS strains isolated from chicken meat in Indonesia. This is the first known report of antimicrobial resistance in NTS strains isolated from Indonesian chicken. The O4 group was the most common serotype of NTS strains, and *S.* Schwarzengrund was mainly detected. *S.* Schwarzengrund is one of the major serotypes isolated from humans and animals and has been reported as an epidemic pathogen in Asia, Denmark and the United States since early 2000 [22]. Among the NTS strains, we observed that 68% (34 of 50 strains) were non-susceptible to antibiotics. Moreso, especially 60% (30 of 50 strains) were non-susceptible to NA. All strains non-susceptible to NA had amino acid mutations at positions 83 and 87, including S83→Y mutation in *gyrA*. NTS strains with S83→Y mutation were also detected in poultry from Brazil [23]. Mutations in QRDR of *gyrA* and/or *parC* genes are most commonly related to the resistance of quinolones in *Salmonella* strains and other bacteria [12]. 

Non-susceptible strains of TC were the second-most detected (54%) because TC was commonly used in poultry feed for protection against infectious diseases and growth promotion. Of 27 non-susceptible strains to TC, 81.5% carried *tetA* and/or *tetB*, two genes most frequently involved in tetracycline resistance in NTS strains. The prevalence of *tetA* was higher than that of *tetB*, consistent with studies in the Nigeria [18,24,25,26].

The non-susceptible rate to ABPC, GM or KM in NTS strains was lower than that of NA or TC, however, the prevalence of *bla*_TEM_ or *aadA* was high. Since *bla*_TEM_ and *aadA* are present on plasmids, they may lead to horizontal transmission of antimicrobial resistance genes across other *Salmonella* or other bacteria. Moreover, strains non-susceptible to antibiotics had a significantly high prevalence of virulence genes, such as *tcfA*, *cdtB*, *fimA* and *sfbA*, encoding ciliary proteins, intracellular survival, adhesion and iron uptake, than susceptible strains. 

Among the NTS strains, *S.* Schwarzengrund was significantly more resistant to ABPC, KM, TC and NA than other serotypes. Of 23 *S*. Schwarzengrund, 22 (95.7%) were non-susceptible to two or more antibiotics, and 11 (47.8%) were non-susceptible to ABPC, KM and/or GM, TC and NA, including CPFX. Furthermore, it has been reported that multidrug-resistant *S.* Schwarzengrund was isolated from food, including chicken, and from human samples in Taiwan, Thailand, Denmark and the United States [27]. We also found that one strain of *S.* Schwarzengrund isolated from a fetal specimen had multidrug resistance in Japan [28]. In addition to the virulence genes above, *S*. Schwarzengrund also had *hlyE* and *phoP/Q* encoding toxin production and survival within macrophages, respectively. The virulence rate of *S*. Schwarzengrund was higher than for other serotypes. *S*. Schwarzengrund strains were approximately identified as ST96. ST96 strains were reported to carry *mcr-1*, a plasmidic gene encoding colistin resistance in Brazilian chicken [29]. The multidrug-resistant *S*. Schwarzengrund might easily spread in the human body and become difficult to treat.

The limitations of this study include the number of subjects (50), and that 46% of 50 strains were *S*. Schwarzengrund. Thus, the sample size was insufficient for an epidemiological survey and there were few serotypes to characterize and compare. Additionally, it is necessary to investigate other mechanisms (other antibiotic-inactivating enzymes or efflux pumps, for instance) related to antibiotic resistance [30]. 

## 4. Materials and Methods

### 4.1. Strains 

We isolated 50 strains of *Salmonella enterica* by the following methods from 60 duct rectal swabs and 60 chicken intestines of meats in 12 traditional markets in Surabaya, Indonesia in 2018 [31]. *Salmonella* strains were isolated according to the methods of the *Bacteriological Analytical Manual* [32], with some modifications. 

### 4.2. Serotyping

The isolates were serotyped with polyvalent O and H antiserum by the agglutination method using the *Salmonella* immune serum “Seiken” (Denka Seiken, Tokyo, Japan). 

### 4.3. Antimicrobial Susceptibility Testing

The test for the NTS strains measured 11 antibiotics (ampicillin: ABPC, amoxicillin/clavulanate: AMPC/CVA, ceftriaxone: CTRX, imipenem: IPM, gentamicin: GM, kanamycin: KM, azithromycin: AZM, tetracycline: TC, ciprofloxacin: CPFX, nalidixic acid: NA, chloramphenicol: CP) by the microdilution method using Optipanel E063 (Kyokuto Pharmaceutical Industrial Co., Ltd., Osaka, Japan). In the Optipanel, 96-well microtiter plates containing cation-adjusted Muller–Hinton broth with twofold dilutions of each antimicrobial solution were prepared according to the Clinical and Laboratory Standards Institute (CLSI) recommendations [33]. The *Escherichia coli* ATCC 25922 strain was used for quality control. Criteria were in accordance with CLSI M100-ED31 [33].

### 4.4. DNA Extraction and Detection of Antimicrobial Resistant Genes

Bacterial DNA was extracted by the boiling method. The bacterial samples were suspended in the Tris-HCL buffer, incubated at 100 °C for 15 min and immediately cooled, centrifuged at 13,000 rpm for 5 min, and the supernatant was collected. Primers were shown in Table 6. We detected antimicrobial resistance genes (*bla*_TEM_ for β-lactam resistance; *aadA* for aminoglycoside resistance; *tetA, tetB, tetC and tetG* for tetracycline resistance) by PCR using TaKaRa Ex Taq (TaKaRa, Shiga, Japan) [34,35,36]. 

We also detected amino acid mutations in QRDR (*gyrA, gyrB, parC* and *parE*), and the plasmid-mediated quinolone resistance (PMQR) genes (*qnrA, qnrB, qnrS, qepA* and *aac (6′)-Ib-cr*) by PCR and sequencing [37,38,39]. The purification of PCR products was conducted with the QIAquick PCR purification kit (QIAGEN, Hilden, Germany) and the sequencing analysis was done by Eurofins Genomics (Eurofins Genomics, Tokyo, Japan).

### 4.5. Detection of Genes Encoding Virulence Factors

Thirteen genes of encoding virulence factors (*invA, sopB, ssel, tcfA, hlyE, cdtB, sfbA, agfA, fimA, fliC, spvC, slyA* and *phoP/Q*) were detected by PCR and sequencing analysis [5,6,7,40]. The PCR conditions were 94 °C, 2 min; 35 cycles of 94 °C, 1 min; 55 °C, 1 min; 72 °C, 2 min; 72 °C, 4 min.

### 4.6. MLST

MLST was conducted by PCR amplification and sequencing of seven housekeeping genes (*suc, hisD, thr, pur, dnaN, hem* and *aro*) [41]. The temperature conditions were initial denaturing at 94 °C for 2 min, followed by 25 cycles of denaturation at 94 °C for 1 min, annealing at 53 °C (*dnaN*) or 60 °C (except *dnaN*) each for 1 min, extension at 72 °C for 1 min and a final extension at 72 °C for 4 min. ST was determined using the MLST website [42].

### 4.7. Statistical Analysis

Significant differences between serotypes and antimicrobial susceptibilities, or serotypes and virulence genes, were determined by Fisher’s exact test using SPSS software, version 24.0 (SPSS, Chicago, IL, USA). *p* < 0.05 was considered statistically significant.

## 5. Conclusions

We found that NTS strains isolated from Indonesian chicken had a high resistance to antibiotics and many virulence factors. In particular, *S*. Schwarzengrund strains belonging to ST96 were the most frequently detected as multi-antimicrobial resistant and had a high prevalence of virulence genes. These NTS strains in food or other environments might be transmitted to humans, and it is necessary to continue investigating NTS strains with resistance to antibiotics.

## Figures and Tables

**Table 1 pathogens-11-00543-t001:** Distribution of O groups and serotypes in 50 non-typhoidal *Salmonella* (NTS) strains.

O Group	Serotype	Strains (%)
O2	*S.* Kiel	1 (2)
	*S.* Nitra	1 (2)
O4	*S.* Schwarzengrund	23 (46)
	*S.* Tokoin	2 (4)
	*S.* Typhimurium	2 (4)
	*S.* Budapest	1 (2)
O7	NT ^※^	3 (6)
O8	*S.* Istanbul	4 (8)
	*S.* Corvallis	2 (4)
	*S.* Portanigra	1 (2)
	*S.* Herston	1 (2)
	NT	3 (6)
O9	*S.* Enteritidis	2 (4)
	NT	1 (2)
O3,10	NT	2 (4)
O1,3,19	*S.* Liverpool	1 (2)
	total	50

^※^ NT: not typed.

**Table 2 pathogens-11-00543-t002:** Antimicrobial susceptibility rates of *S.* Schwarzengrund and other serotypes of NTS strains.

Antibiotic *	Number of Non-Susceptible Strains (%)			*p*-Value #
	Total *n* = 50	*S.* Schwarzengrund *n* = 23	Other Serotypes *n* = 27	
**ABPC**	13 (26)	11 (47.8)	2 (7.4)	**0.001**
**AMPC/CVA**	3 (6)	1 (4.3)	2 (7.4)	1.000
**CTRX**	0	0	0	-
**IPM**	0	0	0	-
**GM**	6 (12)	5 (21.7)	1 (3.7)	0.070
**KM**	8 (16)	7 (30.4)	1 (3.7)	**0.007**
**AZM**	0	0	0	-
**TC**	27 (54)	22 (95.7)	5 (18.5)	**<0.001**
**CPFX**	5 (10)	2 (8.7)	3 (11.1)	1.000
**NA**	30 (60)	23 (100)	7 (25.9)	**<0.001**
**CP**	0	0	0	-

* ABPC: ampicillin, AMPC/CVA: amoxicillin/clavulanate, CTRX: ceftriaxone, IPM: imipenem, GM: gentamicin, KM: kanamycin, AZM: azithromycin, TC: tetracycline, CPFX: ciprofloxacin, NA: nalidixic acid, CP: chloramphenicol. # bold indicates significant levels, *p* < 0.05.

**Table 3 pathogens-11-00543-t003:** Prevalence of antimicrobial resistance genes in *S.* Schwarzengrund and other serotype strains.

Antimicrobial Resistance Gene	Number of Strains (%)		
	Total	*S.* Schwarzengrund	Other Serotypes
** *bla* ** ** _TEM_ **	12	11 (91.7)	1 (8.3)
** *aadA* **	4	4 (100)	0
** *tetA* **	15	11 (73.3)	4 (26.7)
** *tetB* **	6	5 (83.3)	1 (16.7)
** *tetA* ** **and *tetB***	1	1 (100)	0
** *tetC* **	0	0	0
** *tetG* **	0	0	0
**Mutation of *gyrA***	30	23 (76.7)	7 (23.3)
**Mutation of *gyrB***	0	0	0
**Mutation of *parC***	1 *	0	1 (100)
**Mutation of *parE***	0	0	0
** *qnrA* **	0	0	0
** *qnrB* **	0	0	0
** *qnrS* **	0	0	0
** *aac(6′)-lb-cr* **	0	0	0
** *qepA* **	0	0	0

* The strain had also 2 mutations of *gyrA* (S83→F and D87→N).

**Table 4 pathogens-11-00543-t004:** Prevalence of virulence genes in *S.* Schwarzengrund and other serotype strains.

Virulence Gene	Number of Strains (%)			*p*-Value *
	Total *n* = 50	*S.* Schwarzengrund *n* = 23	Other Serotypes *n* = 27	
** *invA* **	50 (100)	23 (100)	27 (100)	-
** *sopB* **	44 (88)	21 (91.3)	23 (85.2)	0.674
** *ssel* **	0	0	0	-
** *tcfA* **	37 (74)	23 (100)	14 (51.9)	<0.001
** *hlyE* **	43 (86)	23 (100)	20 (74.1)	0.011
** *cdtB* **	25 (50)	21 (91.3)	4 (14.8)	<0.001
** *sfbA* **	31 (62)	23 (100)	8 (29.6)	<0.001
** *agfA* **	44 (88)	22 (95.7)	22 (81.5)	0.199
** *fimA* **	32 (64)	21 (91.3)	11 (40.7)	<0.001
** *fliC* **	0	0	0	-
** *spvC* **	0	0	0	-
** *slyA* **	30 (60)	16 (69.6)	14 (51.9)	0.254
** *phoP/Q* **	43 (86)	23 (100)	20 (74.1)	0.011

* bold indicates significant levels, *p* < 0.05.

**Table 5 pathogens-11-00543-t005:** Classification in 23 *S.* Schwarzengrund strains by Multilocus sequence typing (MLST).

Sequence Type	Allelic Profile in MLST							Number of Strains (%) N = 23
	aroC	dnaN	hemD	hisD	purE	sucA	thrA	
**96**	43	47	49	49	41	15	3	22 (95.7)
Not typed	43	47	49	7	41	15	3	1 (4.3)

**Table 6 pathogens-11-00543-t006:** Primer pairs used for the analysis of antimicrobial-resistant genes.

Target Genes	Amplicon Size (bp)	Tm (°C)	Primer	Sequence	Reference
*bla* _TEM_	690	60	*bla*_TEM_F	5′-TTTCGTGTCGCCCTTATTC-3′	[34]
			*bla*_TEM_R	5′-CCGGCTCCAGATTTATCA-3′	
*aadA*	525	60	*aadA*F	5′-GTGGATGGCGGCCTGAA-3′	
			*aadA*R	5′-AATGCCCAGTCGGCAGC-3′	
*tetA*	201	55	*tetA*F	5′-GCTACATCCTGCTTGCCT-3′	
			*tetA*R	5′-CATAGATCGCCGTGAAG-3′	
*tetB*	173	63	TetBGK-F2^m^	5′-CGCCCAGTGCTGTTGTTGTC-3′	[35]
			TetBGK-R2^m^	5′-CGCGTTGAGAAGCTGAGGTG-3′	
*tetC*	505	50	TetCF	5′-GGTTGAAGGCTCTCAAGGGC-3′	[36]
			TetCR	5′-CCTCTTGCGGGAATCGTCC-3′	
*tetG*	662	52	TetGF	5′-GCAGCGAAAGCGTATTTGCG-3′	
			TetGR	5′-TCCGAAAGCTGTCCAAGCAT-3′	
*gyrA*	251	58.6	stgyrA1	5′-CGTTGGTGACGTAATCGGTA-3′	[37]
			stgyrA2	5′-CCGTACCGTCATAGTTATCC-3′	
*gyrB*	181	58	stmgyrB1	5′-GCGCTGTCCGAACTGTACCT-3′	[38]
			stmgyrB2	5′-TGATCAGCGTCGCCACTTCC-3′	
*parC*	270	67	stmparC1	5′-CTATGCGATG TCAGAGCTGG-3′	
			stmparC2	5′-TAACAGCAGCTCGGCGTATT-3′	
*parE*	240		stmparE1	5′TCTCTTCCGATGAAGTGCTG-3′	
			stmparE2	5′-ATACGGTATAGCGGCGGTAG-3′	
*qnrA*	516	53	qnrA F	5′-ATTTCTCACGCCAGGATTTG-3′	[39]
			qnrA R	5′-GATCGGCAAAGGTTAGGTCA-3′	
*qnrB*	469	59	qnrB F	5′-GATCGTGAAAGCCAGAAAGG-3′	
			qnrB R	5′-ACGATGCCTG¬GTAGTTGTCC-3′	
*qnrS*	417		qnrS F	5′-ACGACATTCGTCAACTGCAA-3′	
			qnrS R	5′- TAAATTGGCACCCTGTAGGC-3′	
*aac (6* *′)-Ib-cr*	554		aac(6′)-Ib-cr F	5′-TGACCAACAGCAACGATTCC-3′	
			aac(6′)-Ib-cr R	5′-TTAGGCATCACTGCGTGTTC-3′	
*qepA*	720		qepA F	5′-GGACATCTACGGCTTCTTCG-3′	
			qepA R	5′-AGCTGCAGGTACTGCGTCAT-3′	

## Data Availability

The data sets analyzed in the present study are available from the corresponding author on reasonable request.

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
