# Peer review of "Antibiotic Resistance in Non-Typhoidal Salmonella enterica Strains Isolated from Chicken Meat in Indonesia"

_pathogens, 2022, doi:10.3390/pathogens11050543_

Round 1

Reviewer 1 Report

The manuscript is well written and easy to follow. The study design and results presentation are clear. The only suggestion is from line 190-191, there is no information about how these 50 Salmonella strains were isolated, such as from how many duct rectal swab/ chicken intestine meat samples, and using what kind of method for isolation.

Author Response

Dear  reviewer 1,

Thank you for your comments.

Comments and Suggestions for Authors

The manuscript is well written and easy to follow. The study design and results presentation are clear. The only suggestion is from line 190-191, there is no information about how these 50 Salmonella strains were isolated, such as from how many duct rectal swab/ chicken intestine meat samples, and using what kind of method for isolation.

Response: Thank you for the comments. We add to the section name, sample number and add to the isolation methods of Salmonella strains from samples as follows. We also add to the reference [41].

  1. Materials and Methods

4.1. Strains

We isolated 50 strains of Salmonella enterica by the following methods from 60 duct rectal swabs and 60 chicken intestine of meat in 12 traditional markets in Surabaya, Indonesia in 2018 [41]. Salmonella strains were isolated according to the methods of Bacteriological Analytical Manual (2004) with some modifications. Twenty-five gram of each sample was chopped aseptically and homogenized with 225mL of Buffered Pepton Water (BPW, Oxoid, Ontario, Canada). Subsequently, the  pre-enrichment was incubated at 37°C for 24hours. One ml of BPW aseptically added into 10mL Selenite Cystine Broth (SCB, Oxoid), incubated at 37°C for 24 hours. A loop full of the selectively enriched suspension was streaked onto Xylose Ly-sine Deoxycholate (XLD, Oxoid) agar and incubated at 37°C for 24 hours. A single colony grew on the XLD agar was observed with the specific criteria of Salmonella spp. The criteria for confirmation of Salmonella isolates were based on microscopic examination with Gram staining and biochemical profiling. The colonies with gram-negative and short-rod were evaluated by biochemical tests (IMViC test). The TSIA (Triple Sugar Iron Agar) test was conducted to check for their ability to ferment glucose, lactose and sucrose sugars, gas and H2S production. The IMViC tests also for Indole production, Methyl Red, Voges-Proskauer, and the use of Simmon’s citrate utilization. The Salmonella isolates were stored at -80°C in Luria-Bertani (LB, Difco) broth with the addition of 30% glycerol (v/v).

(lines 192 to 210)

Best  regards,

Kayo Osawa, PhD

Kobe Tokiwa University

Reviewer 2 Report

  • Line 62, correct:’ …. , tetA, tetB, tetC and tetG: tetracycline resistance.
  • In line 73, you wrote ‘..and antimicrobial susceptibilities of clinical NTS isolated from chicken meat in Indonesia’ delete ‘clinical’ since your isolates are from chicken meat, not from patients or from ill chickens.
  • Line 196, You wrote ‘..The test for the NTS strains measured 11 antibiotics (ABPC, AMPC/CVA, CTRX, Imipenem: IPM, GM, KM, AZM, TC, CPFX, NA, Chloramphenicol: CP)’ . PLEASE WRITE THE NAME OF THE all ANTIBIOTICS THEN the abbreviation of the antibiotics. ABPC???, CTRX???. For example ’…imipenem (IMP), gentamicin (GM), Kanamycin (KM)….’
  • Line 202, provide de detail of DNA extraction method, what you wrote is not clear
  • Line 206, write :’.. and the plasmid-mediated quinolone resistance (PMQR) genes (qnrA, qnrB, qnrS, qepA and aac (6’)-Ib-cr) by PCR and sequencing’
  • Line 2012, instead of ‘4.5. Detection of pathogenic genes’ write ‘4.5. Detection of genes encoding virulence factors’ we can nor call a gene ‘pathogenic’, bacteria can be pathogenic but a gene is a locus that encode virulence factor or resistance to antibiotic or other products. So write also ‘Thirteen genes encoding virulence factors (…….’
  • Write ‘PCR conditions were as follows : 94°C 2 min ; 35 cycles of 94°C 1 min, 55°C 1 min, 72°C 2 min; 72°C 4 min.’
  • Please verify, the Figure 1was made according to the table 3 (Table 3. Prevalence of virulence genes in S. Schwarzengrund and other serotype strains) or according to susceptible and non susceptible strains. I believe that the figure 1 and this analyse is not necessary, in reality you compare S. Schwarzengrund with other serotypes which is already presented in Table3. Perhaps this will be important if for example you analyse S. Schwarzengrund containing susceptible isolates and resistant isolates. Really I believe that figure 1 is the same as the table 3. So this association is linked to the serovar Schwarzengrund not to antibiotic resistance.
  • Line 156, please correct and write ‘Mutations in QRDR of gyr A and/or parC genes are most commonly related with resistance to quinolones in Salmonella strains and other bacteria’
  • Line 179, Write : ST96 strains were reported to carry mcr-1, a plasmidic gene encoding colistin resistance in Brazilian chicken [39]. The multidrug-resistant S. Schwarzengrund might easily spread in the human body and become difficult to treat.

Author Response

Dear Reviewer 2,

Response: We are extremely grateful for many comments. We corrected according to the comments as follows.

Line 62, correct:’ …. , tetA, tetB, tetC and tetG: tetracycline resistance.

Response: Thank you for the comment. We correct it.

  1. Introduction

・・antimicrobial efflux (qepA: quinolone resistance, tetA, tetB, tetC and tetG: tetracycline resistance), and restriction of antimicrobial uptake [12, 14, 15].

(lines 61 to 63).

In line 73, you wrote ‘..and antimicrobial susceptibilities of clinical NTS isolated from chicken meat in Indonesia’ delete ‘clinical’ since your isolates are from chicken meat, not from patients or from ill chickens.

Response: Thank you for the comment. We deleted the word ‘clinical’.

  1. Introduction

In this study, we confirmed the virulence genes and antimicrobial susceptibilities of NTS isolated from chicken meat in Indonesia, and also detected antimicrobial-resistant genes.

(lines 76 to 78)

Line 196, You wrote ‘..The test for the NTS strains measured 11 antibiotics (ABPC, AMPC/CVA, CTRX, Imipenem: IPM, GM, KM, AZM, TC, CPFX, NA, Chloramphenicol: CP)’ . PLEASE WRITE THE NAME OF THE all ANTIBIOTICS THEN the abbreviation of the antibiotics. ABPC???, CTRX???. For example ’…imipenem (IMP), gentamicin (GM), Kanamycin (KM)….’

Response: Thank you for the comment. We added the information of antibiotics.

4.3. Antimicrobial susceptibility testing

The test for the NTS strains measured 11 antibiotics (Ampicillin: ABPC, Amoxicil-lin/Clavulanate: AMPC/CVA, Ceftriaxone: CTRX, Imipenem: IPM, Gentamicin: GM, Kanamycin: KM, Azithromycin: AZM, Tetracycline: TC, Ciprofloxacin: CPFX, Nalidixic acid: NA, Chloramphenicol: CP) by the microdilution method using Optipanel E063 (Kyokuto Pharmaceutical Industrial Co., Ltd, Osaka, Japan).

(lines 215 to 219)

Line 202, provide de detail of DNA extraction method, what you wrote is not clear

Response: Thank you for the comment. We added the DNA extraction method in detail.

4.4. DNA extraction and detection of antimicrobial resistant genes

Bacterial DNA was extracted by the boiling method. The bacterial samples were sus-pended in the Tris-HCL buffer, incubated at 100°C for 15 min and immediately cooled, centrifuged at 13,000 rpm for 5 min, and collected the supernatant. Primers were shown in Table 5. We detected antimicrobial resistance genes・・・

(lines 223 to 229)

Line 206, write :’.. and the plasmid-mediated quinolone resistance (PMQR) genes (qnrA, qnrB, qnrS, qepA and aac (6’)-Ib-cr) by PCR and sequencing’

Response: Thank you for the comment. We correct it.

4.4. DNA extraction and detection of antimicrobial resistant genes

We also detected amino acid mutations in QRDR (gyrA, gyrB, parC and parE), and the plasmid-mediated quinolone resistance (PMQR) genes (qnrA, qnrB, qnrS, qepA and aac (6’)-Ib-cr) by PCR and sequencing [24-26].

(lines 231 to 233)

Line 2012, instead of ‘4.5. Detection of pathogenic genes’ write ‘4.5. Detection of genes encoding virulence factors’ we can nor call a gene ‘pathogenic’, bacteria can be pathogenic but a gene is a locus that encode virulence factor or resistance to antibiotic or other products. So write also ‘Thirteen genes encoding virulence factors (…….’

Response: Thank you for the comment. We revise the words.

Abstract:

In conclusion, NTS strains isolated from Indonesian chicken had high resistance to antibiotics and many virulence factors.

(line 34)

Keywords: Non-typhoidal Salmonella enterica (NTS); Indonesia; antimicrobial resistance; viru-lence factors; chicken

(lines 37 to 38)

2.4. Detection of genes encoding virulence factors

(line 130)

Moreover, strains non-susceptible to antibiotics had significantly high prevalence of virulence genes such as tcfA, cdtB, fimA, and sfbA, encoding ciliary proteins, intracellular survival, adhesion and iron uptake, than susceptible strains.

(line 172 to 175)

4.5. Detection of genes encoding virulence factors

Thirteen genes of encoding virulence factors (invA , sopB, ssel, tcfA, hlyE, cdtB, sfbA, agfA, fimA, fliC, spvC, slyA and phoP/Q) were detected by PCR and sequencing analysis

(line 237 to 238)

  1. Conclusions

We found that NTS strains isolated from Indonesian chicken had high resistance to antibiotics and many virulence factors.

(lines 52 to 53)

Write ‘PCR conditions were as follows : 94°C 2 min ; 35 cycles of 94°C 1 min, 55°C 1 min, 72°C 2 min; 72°C 4 min.’

Response: Thank you for the comment. We correct it.

PCR conditions were 94°C 2 min; 35 cycles of 94°C 1 min, 55°C 1 min, 72°C 2 min; 72°C 4 min.

(lines 239 to 240)

Please verify, the Figure 1was made according to the table 3 (Table 3. Prevalence of virulence genes in S. Schwarzengrund and other serotype strains) or according to susceptible and non susceptible strains. I believe that the figure 1 and this analyse is not necessary, in reality you compare S. Schwarzengrund with other serotypes which is already presented in Table3. Perhaps this will be important if for example you analyse S. Schwarzengrund containing susceptible isolates and resistant isolates. Really I believe that figure 1 is the same as the table 3. So this association is linked to the serovar Schwarzengrund not to antibiotic resistance.

Response: Thank you for the comment. We agree and delete Figure 1.

Line 156, please correct and write ‘Mutations in QRDR of gyr A and/or parC genes are most commonly related with resistance to quinolones in Salmonella strains and other bacteria’

Response: Thank you for the comment. We correct it.

Mutations in QRDR of gyrA and/or parC genes are most commonly related with resistance to quinolones in Salmonella strains and other bacteria [12].

(lines 161 to 163)

Line 179, Write : ST96 strains were reported to carry mcr-1, a plasmidic gene encoding colistin resistance in Brazilian chicken [39]. The multidrug-resistant S. Schwarzengrund might easily spread in the human body and become difficult to treat.

Response: Thank you for the comment. We correct it.

ST96 strains were reported to carry mcr-1, a plasmidic gene encoding colistin resistance in Brazilian chicken [39]. The multidrug-resistant S. Schwarzengrund might easily spread in the human body and become difficult to treat.

(lines 184 to 185)

Best regards,

Kayo Osawa, PhD

Kobe Tokiwa University

Reviewer 3 Report

The manuscript “Antibiotic resistance in non-typhoidal Salmonella enterica 2 strains isolated from chicken meat in Indonesia” by Minori Takaichi et al, describes the presence of antimicrobial and virulence genes in a collection of 50 Non Thyphodal Salmonella isolated from chicken meat in Indonesia. The analysis of the data could be more detailed, and the discussion of the results is incipient. Also, no further development to the field is achieved by this study.

Line 26: TC- Use the full name of the antimicrobial since in it is the abstract

Line 88: 2.2. Antimicrobial susceptibility testing

Other results could be obtained from this data, as the % of multidrug resistance of the isolates.

Line 107: 2.3. Detection of antimicrobial resistance genes   instead of 2.3. Detection of antimicrobial genes.

              In this section you could use the data to produce a graphic to make the results easier to follow.  

Line 140- 2.5. MLST

From the MLST data a phylogenetic tree could be produced. Also, data should be presented as supplementary material.

Author Response

Dear reviewer 3,

Thank you for your comments.

Comments and Suggestions for Authors

The manuscript “Antibiotic resistance in non-typhoidal Salmonella enterica 2 strains isolated from chicken meat in Indonesia” by Minori Takaichi et al, describes the presence of antimicrobial and virulence genes in a collection of 50 Non Thyphodal Salmonella isolated from chicken meat in Indonesia. The analysis of the data could be more detailed, and the discussion of the results is incipient. Also, no further development to the field is achieved by this study.

Line 26: TC- Use the full name of the antimicrobial since in it is the abstract

Response: Thank you for the comment. We revise it.

Abstract:

Among 25 tetracycline (TC) non-susceptible strains, 22 (88%) had tetA or tetB. The non-susceptibility rates to Ampicillin, Gentamicin or Kanamycin were lower than that of NA or TC, but the prevalence of blaTEM or aadA were high.

(lines 26 to 28)

Line 88: 2.2. Antimicrobial susceptibility testing

Other results could be obtained from this data, as the % of multidrug resistance of the isolates.

Response: Thank you for the comment. We add the following data.

2.2. Antimicrobial susceptibility testing

Of 34 strains, 27 strains (79.4%) were non-susceptible to more than one antibiotic. Thirteen strains (38.2%) were non-susceptible to TC and NA including CPFX, and 11 strains (32.4%) were to ABPC, KM and /or GM, TC, and NA including CPFX.

(line 106 to 109)

Line 107: 2.3. Detection of antimicrobial resistance genes   instead of 2.3. Detection of antimicrobial genes.

Response: Thank you for the comment. We correct it.

2.3. Detection of antimicrobial resistance genes

(line 110)

In this section you could use the data to produce a graphic to make the results easier to follow. 

Response: Thank you for the comment. We add the following data and Table 3.

2.3. Detection of antimicrobial resistance genes

The prevalence of antimicrobial resistance genes were shown in Table 3. Most of non-susceptible strains to ABPC (12 of 13 strains, 92.3%) possessed blaTEM. 11 strains were S. Schwarzengrund and 1 strain was S. Budapest. Among 6 non-susceptible strains to GM, 4 (66.7%) had aadA and all of them were S. Schwarzengrund. In 27 non-susceptible strains to TC, 22 (81.5%) had tetA or tetB, of which 15 (55.6%) had tetA and 7 (25.9%) had tetB, but not tetC or tetG. Of 15 strains with tetA, 11 were S. Schwarzengrund and 4 were S. Istanbul. Of 7 strains with tetB, 6 were S. Schwarzengrund and 1 was S. Budapest. Among 30 strains with non-susceptible to NA, QRDR mutations, especially gyrA, were detected in all strains. The S83→Y mutation in gyrA was the most common mutation (27 strains; 90%) including all 23 strains of S. Schwarzengrund. The D87→N mutation in gyrA was found in 3 strains (10%) including 2 strains of S. Enteritidis. The S83→F mutation in gyrA was found in 1 strain (3.3%), which belonged to the O8 group but could not be serotyped. In addition, only 1 strain (3.3%) had a mutation in parC (S81→I) with 2 mutations of gyrA (S83→F and D87→N). PMQR was not found in these strains.

(lines 116 to 129)

Line 140- 2.5. MLST

From the MLST data a phylogenetic tree could be produced. Also, data should be presented as supplementary material.

Response: Thank you for the comment. In this study, We found the distinct difference between S. Schwarzengrund and other serotypes in the rates of resistance to antimicrobial agents and in the carriage of antimicrobial resistance genes and virulence genes. Therefore, we conducted MLST focusing on S. Schwarzengrund and did not detected the MLST data in the strains of other serotypes.

Best regards,

Kayo Osawa, PhD

Kobe Tokiwa University

Round 2

Reviewer 3 Report

Dear authors, 

I acknowledge changes made, which I believe improved the manuscript, however I believe that the Discussion session should still be improved, as it wasn’t significantly changed in the last revision.

Author Response

Dear reviewer 3,

It is our great pleasure to have this opportunity to revise our article again.

Sincerely, 

Kayo Osawa,  PhD

Kobe Tokiwa University
